A dual nonsubsampled contourlet network for synthesis images and infrared thermal images denoising

Xu Zhendong 1
Zhao Hongdan 1
Zheng Yu 1
Guo Hongbo 1
Li Shengyang 1
Lyu Zhiyu 2 lyuzhiyu1216@163.com
1 State Grib Jilin Electric Power Co., Ltd, Liaoyuan Power Supply Company , Liaoyuan , China
2 School of Automation Engineering, Northeast Electric Power University , Jilin , China
Angiulli Giovanni
Electronic publication date: 2024 Jan 26
Publication date: 2024
Volume: 10
Electronic Location ID: e1817
Received 2023 Oct 9; Accepted 2023 Dec 19
Copyright: © 2024 Xu et al.
Copyright year: 2024
Copyright holder: Xu et al.
License: This is an open access article distributed under the terms of the Creative Commons Attribution License, which permits unrestricted use, distribution, reproduction and adaptation in any medium and for any purpose provided that it is properly attributed. For attribution, the original author(s), title, publication source (PeerJ Computer Science) and either DOI or URL of the article must be cited.
License URL: https://creativecommons.org/licenses/by/4.0/

Keywords: Nonsubsampled contourlet, Image denoising, CNN

Funding: State Grid Jilin Electric Power Co., LTD. Technology 2023-23 This work was supported by State Grid Jilin Electric Power Co., LTD. Technology project (No. 2023-23). The funders had no role in study design, data collection and analysis, decision to publish, or preparation of the manuscript.

==============================
The most direct way to find the electrical switchgear fault is to use infrared thermal imaging technology for temperature measurement. However, infrared thermal imaging images are usually polluted by noise, and there are problems such as low contrast and blurred edges. To solve these problems, this article proposes a dual convolutional neural network model based on nonsubsampled contourlet transform (NSCT). First, the overall structure of the model is made wider by combining the two networks. Compared with the deeper convolutional neural network, the dual convolutional neural network (CNN) improves the denoising performance without increasing the computational cost too much. Secondly, the model uses NSCT and inverse NSCT to obtain more texture information and avoid the gridding effect. It achieves a good balance between noise reduction performance and detail retention. A large number of simulation experiments show that the model has the ability to deal with synthetic noise and real noise, which has high practical value.

Introduction

A fire accident caused by the failure of electrical equipment will cause a lot of property losses; therefore, the safe operation of electrical equipment has attracted more and more attention, which is also a problem that must be solved in the actual production (Wang et al., 2021; Zhou et al., 2023). As an important item of electrical equipment in the power system, the electrical switchgear is often used to realize the transmission and exchange of power loads. In the event of failure, the electrical switch cabinet can exit the faulty equipment and line segments in the power system in a timely and effective manner, thus ensuring the safe and stable operation of the power system and production equipment. For discerning the fault of a switch cabinet (Zhang et al., 2020), the most direct way is to use infrared thermal imaging technology for temperature measurement (Li et al., 2021; Thukral et al., 2022). However, infrared thermal imaging images are usually polluted by noise, and there are problems such as low contrast and blurred edges. Therefore, image denoising is very important.

With the development of deep learning (Xu et al., 2023; Mi et al., 2021; Ma et al., 2021), its powerful modeling ability and network training improved with the iteration of hardware devices have made it achieve great success in related fields of image processing, and the research focus of image denoising methods has gradually shifted to methods based on deep learning (Tian et al., 2023; Zhang et al., 2022). However, in the actual image denoising and restoration tasks, the existing models do not solve the following three problems:

1) In most cases, the better the denoising effect of the model, the more serious the loss of image detail information, and the balance between the two is difficult to achieve.

2) For better denoising effect, more features need to be learned, which requires a deeper network and a larger receptive field. However, whether the network is deepened or the traditional method of expanding receptive field, the training efficiency will decrease significantly, and problems such as gradient disappearance or overfitting will occur.

3) Most noise removal methods are designed to remove additive white Gaussian noise (AWGN), and the noise type and noise level are set in advance, while some prior knowledge of the real noise is unknown. Therefore, few studies have proposed methods to remove real noise.

To solve these problems, this article proposes a new dual convolutional network model, named NCTBNet, which combines nonsubsampled contourlet transform (NSCT) and convolutional neural networks (CNNs). First, the model combines two U-Net in parallel to increase the width of the network. Compared with deepening the network, increasing the width of the network can obtain more texture and details, reduce the computational cost and effectively avoid overfitting. Secondly, in order to avoid information loss caused by upsampling and downsampling operations in U-net, our model uses multi-scale geometric transformation to replace the pooling layer and deconvolution layer, which has been proved to be effective in solving grid effects. NSCT, as one of the classical methods of multi-scale geometric transformation, is used in this model. With the characteristics of multi-resolution, multi-direction and translation invariant, it can effectively extract the edge texture of the image, and is widely used in image fusion and image segmentation. Therefore, we choose to use NSCT and inverse NSCT to decomposition and reconstruction of the image to ensure that the image details can be effectively preserved while denoising.

In summary, the NCTBNet contributions proposed in this article can be summarized as follows:

1) A new dual convolutional neural network model is proposed, which obtains more image textures and details through the parallel network structure, and improves the denoising performance without increasing the computational cost.

2) NSCT and inverse NSCT are used for image decomposition and reconstruction to avoid grid effect and loss of image detail information.

3) A noise map is used as input to make the model more flexible in dealing with realistic noise.

The structure of this article is as follows: In “Introduction”, the basic principle and function of non-subsampled contour wave transform are given; “Materials and Methods” briefly reviews deep learning based denoising methods. In “The Proposed NCTBNet”, the convolutional neural network model is introduced in detail. In “Results”, the experimental methods, experimental procedures and comparative experiments are described in detail. “Conclusions” summarizes the full work.

Materials and Methods

Deep learning has been widely used in the field of image processing, and has made a qualitative leap in various visual tasks, especially in the field of image denoising, which has become a research hotspot (Dong et al., 2022; Liu et al., 2022; Jiang et al., 2022). DnCNN proves that deep convolutional neural networks have a strong ability to process image noise (Zhang et al., 2017). Then, on the basis of the improvement, a faster and more adaptive noise reduction network model (FFDNet) was proposed (Zhang, Zuo & Zhang, 2018). Goyal et al. (2022) proposed a twin convolutional neural network denoising model to remove noise in medical images. The model uses NSCT to extract features from noise source images, and then uses twin convolutional neural networks to carry out weighted fusion of features (Goyal et al., 2022). Guo et al. (2019) proposed a Convolutional Blind Deno-ising Network (CBDNet) model based on convolutional neural networks. Tian et al. (2021) proposed a dual CNN for realistic image denoising and achieved excellent denoising performance. Scetbon, Elad & Milanfar (2021) combined K-SVD and deep CNN to denoise images and get good denoising performance with small number of parameters. Ghosh et al. (2023) uses LSTM network to remove artifacts from signals, which is also applicable to thermal image denoising. Chen et al. (2023) used a nonconvex low rank model and TV regularization to denoise images and got good denoising performance.

The proposed NCTBNet

Nonsubsampled contourlet transform

The non-subsampled contourlet transform is an improvement of the Contourlet transform (Da Cunha, Zhou & Do, 2006). It is composed of a non-subsampled pyramid and a non-subsampled direction filter bank. It has multi-directional characteristics and good time-frequency characteristics, and can retain the contour information of the image edge well. NSCT first uses non-subsampled pyramid (NSP) to complete the multi-scale decomposition of the image, and then uses NSDFB to realize the multi-direction decomposition of the image. Different from Contourlet Transform, NSCT eliminates the up-sampling and down-sampling operations in the process of image decomposition and reconstruction, so it has good translation invariance. The NSCT process is shown in Fig. 1A, and the frequency domain division of the subband is shown in Fig. 1B.

Figure 1 Nonsubsampled contourlet transform: (A) NSFB structure and (B) idealized frequency partitioning.

The core structure of NSCT is a two-dimensional dual-channel Non-subsampled filter bank. Figure 2 shows two two-dimensional dual-channel NSFBS used in NSCT. Take the pyramid-shaped NSFBS in Fig. 2A as an example, where the images represent a low-pass subband image and high-pass subband image, respectively, which are low-pass decomposition filters and high-pass decomposition filters. For the low-pass reconstruction filter and for the high-pass reconstruction filter, NSP decomposes the original image into subband images of different scales through the pyramid two-channel decomposition filter, and then reconstructs the subband image into the original image through the two-channel reconstruction filter, which satisfies the Bezout identity, as shown in Eq. (1):

Figure 2 Two-channel NSFBs used in the NSCT.

(A) Pyramid NSFB (B) fan NSFB.

(1) H0G0+H1G1=1

where H0 and H1 denote low-pass subband and high-pass subband, G0 and G1 are corresponding filters.

This allows the image to be rebuilt without errors, which can lead to loss of information. Similarly, NSDFB also uses a fan-shaped two-channel decomposition filter to decompose the original image into subband images in different directions, and then reconstructs the subband image into the original image through a two-channel reconstruction filter.

Since NSCT decomposed most of the smooth region of an image into low-frequency subbands, and the noise and part of texture information into high-frequency subbands, the NSCT-based denoising algorithm mainly uses different classifiers such as CNNs or support vector machine (SVM) to divide the high-frequency coefficients into texture class and noise class. Then the soft threshold algorithm is used to shrink the noise coefficients. The setting of threshold depends on the noise level of texture and image. However, texture and noise levels are at odds with each other, with textures requiring smaller thresholds and high levels of noise requiring the use of larger thresholds to deal with. Therefore, the soft threshold algorithm cannot completely remove noise and often makes the de-noised image too smooth. In this article, we design a novel CNN model by using NSCT and inverse NSCT to replace the pooling layer and the upper sampling layer. Since NSCT has translation invariance, multi-scale and multi-direction, it will not cause image information loss in the process of image decomposition and reconstruction, which can effectively solve the lack of correlation of pixels obtained after deconvolution of the upper sampling layer. This can lead to grid effects, loss of image details and other problems. Figure 3 shows the receptive field after deconvolution and the receptive field after inverse NSCT. It is obvious that inverse NSCT can well solve the problems of sparse sampling and local information irrelevance caused by upsampling operation, thus improving the performance of the model.

Figure 3 The illustration of the gridding effect.

(A) The deconvolution layer; (B) the inverse NSCT layer.

The architecture of NCTBNet

The structure of the NCTBNet model proposed in this article is shown in Fig. 4. The model includes two U-Nets. The two U-net models have no difference in the overall structure, and both contain three fully convolutional network (FCN) modules after removing the pooling layer. The first module and the third module consist of four layers of CNNS, and the second module contains eight layers of CNNS.

Figure 4 The architecture of the proposed network.

Starfish image source credit: Martin et al. (2001).

Initially, the U-net model uses pooling layer and deconvolution layer to perform downsampling and upsampling operations, which will cause the loss of image details and affect the denoising result. Therefore, we remove the pooling layer in the model, cancel the up-sampling operation, and replace the up-sampling and down-sampling operation with NSCT and inverse NSCT. As shown in the Fig. 4 the yellow arrow indicates that the image is decomposed into many sub-images by NSCT, and the blue arrow indicates that these sub-images are reconstructed into the original image by inverse NSCT. This ensures that the details of the image are not lost.

In structure, the upper and lower two networks are the same, both are composed of a convolutional layer, batch normalization and rectified linear unit. However, the functions of the two networks are different. The upper network is mainly used to balance the weight between the prior of image noise and the observed value. When the noise density is high, the prior should be given more weight; when the noise density is low, the observed value should be given more weight. The down-network mainly predicts the relevant information of the real noisy image with unknown noise density and noise distribution by sampling each pixel. In terms of setting the size of convolution kernel, current research generally believes that the convolution kernel with the size of 3 * 3 takes into account both performance and efficiency. Therefore, we set the size of convolution kernel to 3 * 3 * c, where c represents the number of channels, and c = 1 when processing grayscale images and c = 3 when processing color images. Considering that the color image requires more feature maps, we set the number of feature maps of the grayscale image to 64 and the number of feature maps of the color image to 128.

In regard to the role of NSCT in NCTBNet, we have already introduced one of them, that is, using NSCT and inverse NSCT to replace the pooling layer and deconvolution layer in the U-net model, so as to preserve image details and avoid grid effect. On the other hand, it can be seen from Fig. 4 that, in addition to the U-net model, we used NSCT again as a means to increase the number of training samples. As we all know, in the training stage of network models, data augmentation methods are usually used to improve the performance of the models. Commonly used data augmentation methods, such as flipping and rotating the input image to obtain image blocks, increase the number of training samples, but this makes the model can only learn part of the noise in the whole image. If the image contains spatial variation noise, the denoising effect of the network model will be greatly reduced. Therefore, we use NSCT to decompose the original image into several sub-band images, which will not lose the overall noise information while increasing the number of training samples, so that the model can learn more features and noise distribution, so as to improve the model’s processing ability of spatial variation noise and real noise. In summary, the advantages of NSCT are summarized as follows:

1) NSCT can optimize the network structure and widen the network by using parallel structure to reduce the computational complexity and improve the network training efficiency.

2) NSCT and inverse NSCT can be used to increase the number of training samples and retain details while removing noise.

Referring to the training of DnCNN and FFDNet, we choose to use the mean square error as a loss function to optimize the parameters of the network, 𝖰 is the parameters of the NCTBNet, F(yi,𝖰) is the output, x and y denote a ground-truth image and a noisy image respectively, the training dataset is {xi,yi}i=1N. The loss function can be described as:

(2) L(Q)=12Na°‖F(yi,𝖰)−xi‖i=1N2

The ADAM algorithm (Kingma & Ba, 2014) was employed to train the NCTBNet. The learning rate begins from 10−3 to 10−4, when the training error remains constant over 10 sequential epochs, we change the learning rate to 10−6 for another 80 epochs.

Results

Dataset

In order to demonstrate NCTBNet’s performance in removing AWGN and denoising real and thermal infrared images, we separated these two tasks: the AWGN denoising task and realistic denoising task. For AWGN denoising task, we choose BM3D (Dabov et al., 2007), WNNM (Gu et al., 2014), TNRD (Chen & Pock, 2016), DnCNN (Zhang et al., 2017), N2V (Krull, Buchholz & Jug, 2019), GCDN (Valsesia, Fracastoro & Magli, 2020), and BUIFD (El Helou & Süsstrunk, 2020) as comparison methods. The training dataset contains 5,359 images. Also referring to the training dataset of the above model, we selected 3,859 images from the Waterloo Exploration dataset and 1,500 images from the ImageNet data set. The noise level of AWGN was set to sigma = 15, 25, 50 respectively.

At the same time, considering that if the sub-image block is too large for the receptive field, the model will bear huge computational costs; if the sub-image block is too small for the receptive field, the model cannot fully learn features from the noisy image, we randomly crop the training dataset and set the size of the subimage block to 180 × 180. To distinguish between the two models, we will name the model with AWGN removed NSTBNet-S.

Because real noise is more complex than AWGN, and both the noise distribution and noise level are unknown, removing real noise is not the same as removing AWGN. The common method to remove real noise is to estimate the noise level graph first, and then convert the real image denoising to AWGN denoising. However, it is often impossible to accurately estimate the noise level in most cases, which results in the performance of the denoising model is greatly reduced when processing real noise images. At present, most deep learning-based denoising methods need to use pairs of clean images and AWGN with known specific noise level to train the model, which is not suitable for training the denoising model of real noisy images. However, Lehtinen et al. (2018) demonstrated that training a denoising model with only real noise images but no clean image data can make the model’s performance approximate or even exceed that of a model using pairs of clean images and noisy images as training datasets. Therefore, this article uses the noise image as the training sample to train the network model. As with NCTBNet-S, we randomly cropped the training dataset, also setting the size of the sub-image blocks to 180 × 180, and named the model used for real denoising NCTBNet-B.

Testset

In order to display the denoising effect of the model in a more detailed way, we carried out denoising experiments of gray image, real image and thermal infrared image respectively. The test datasets for gray image denoising include Set12 and BSD68, while the test dataset for real image denoising is RENIOR dataset. The images in RENIOR dataset contain unknown distribution and level of noise, and there is no corresponding clean image. The thermal infrared image is derived from the thermal infrared switch cabinet image taken by the field.

Since all objective evaluation indicators need to be calculated according to the ground truth of the image, and the real noise image has no ground truth, it is impossible to evaluate the denoising performance of the real noise image by objective evaluation indicators. Therefore, for the experimental results of removing AWGN, we adopted an objective evaluation index, peak signal-to-noise ratio (PSNR) (Wang et al., 2004), to evaluate the denoising effect, the mathematical representation of PSNR is shown in Eq. (3). As for the denoising experiment results of real noise images, we use subjective evaluation to evaluate the denoising effect. In this article, 50 people were invited to score the denoising effect of real noise images, and the scoring criteria are five points excellent, four points good, three points average, two points poor and one point very poor, and then the scoring results were statistically analyzed.

(3) PSNR=10log⁡[Xmax21MN∑i=1M∑j=1N‖Y(i,j)−X(i,j)‖2]

where Y is the observation value of noisy images, X is the ground truth of images.

Gray-scale images denoising

This section shows the denoising performance of the proposed algorithm for gray composite images. Comparing NCTBNet with several representative image denoising methods, including two denoising methods based on non-local self-similarity BM3D, WNNM, and five deep learn-based denoising models DnCNN, BRDNet (Tian, Xu & Zuo, 2020), N2V, GCDN and BUIFD. Three kinds of noise intensity are used to evaluate the denoising performance of gray-scale images. The denoising results of gray-scale images with noise density of 25 and 50 are shown in Figs. 5 and 6. Table 1 shows the average PSNR of all test samples in the SET12 and BSD68 datasets.

Figure 5 Denoising performance with noise level 25.

(A) Raw image; (B) noisy image; (C) BM3D; (D) DnCNN; (E) BRDNet; (F) NCTBNet-S. Zebra image source credit: Martin et al. (2001).

Figure 6 Denoising performance with noise level 50.

(A) Raw image; (B) noisy image; (C) BM3D; (D) WNNM; (E) TNRD; (F) DnCNN; (G) BRDNet; (H) NCTBNet -S. Ship image source credit: Martin et al. (2001).

Table 1 The PSNR of Set12 and BSD68 (dB).

	σ	BM3D	TNRD	DnCNN	BRDNet	N2V	GCDN	BUIFD	NCTBNet-B	NCTBNet-S	
Set12	15	32.37	32.50	32.86	33.03	26.12	33.14	–	33.08	33.16	
25	29.97	30.05	30.44	30.61	25.01	30.78	–	30.65	30.81	
50	26.72	26.82	27.18	27.45	–	27.60	–	27.44	27.63	
BSD68	15	31.07	31.37	31.73	31.79	25.77	31.82	31.40	31.63	31.79	
25	28.57	28.92	29.23	29.29	24.83	29.35	28.75	29.21	29.39	
50	25.62	25.97	26.23	26.36	–	26.38	25.11	26.23	26.43	
Note:

Bold values indicate the best results.

It can be seen from Table 1 that the denoising performance of the proposed model is superior to other comparison models. We note that the gap between the denoising performance of NCTBNet-S and other comparison methods is greater than 15 noise intensifications at 50 noise intensifications. It is precisely because NSCT decomposition is used to classify the noise and texture in high-frequency sub-images, so they can be effectively separated at high noise levels, thus improving the denoising performance. In particular, the NCTBNet-B is trained without prior knowledge of noise and still has a higher PSNR result than those compared models which using a noise-free image pair trained.

Figures 5 and 6 show the denoising effect of the gray-scale image when the noise level is 25 and 50. It can be seen that some denoisers over-smooth the texture area, such as BM3M and WNNM, and that some denoisers produce fake artifacts, such as TNRD, DnCNN, and BRDNet. Almost all models have a certain blur at high noise intensity. The proposed model not only preserves texture details, but also has satisfactory subjective visual effects in the homogeneous areas.

Realistic images and infrared thermal images denoising

In order to verify the denoising performance of the proposed model on real noise images and thermal infrared images, we used the RENOIR dataset and the thermal infrared images of the switch cabinet collected in the field for testing, and TSWC (Xu, Zhang & Zhang, 2018), CDnCNN and CBDNet (Guo et al., 2019) were used as comparison denoisers. Because the noise type and noise intensity of realistic images are unknown, it is impossible to evaluate them objectively by PSNR. We sought 50 people to rate each denoising performance as a numerical analysis. For statistical purposes, all scores were integer, range and evenly divided, as shown in Table 2. The denoising effect is shown in Figs. 7 and 8.

Table 2 The score of realistic noisy images in RENIOR datasets (average/min-max).

	‘Woman’	‘Singer’	‘Boy’	‘Frog’	‘Dog’	
TWSC	2.6/ (2.0–4.0)	2.7/(2.0–4.0)	4.2/(4.0–5.0)	4.3/(4.0–5.0)	2.3/(2.0–3.0)	
DnCNN	3.6/(3.0–5.0)	2.9/(2.0–4.0)	4.3/(4.0–5.0)	4.3/(4.0–5.0)	3.3/(2.0–4.0)	
CBDNet	3.4/(3.0–4.0)	4.1/(3.0–5.0)	4.1/(3.0–5.0)	4.5/(4.0–5.0)	3.4(3.0–4.0)	
NCTBNet-B	4.3/(4.0–5.0)	4.5/(4.0–5.0)	4.8/(4.0–5.0)	4.8/(4.0–5.0)	4.7/(4.0–5.0)	
Note:

Results for NCTBNet-B are shown in bold.

Figure 7 The denoising results of realistic images.

From left to right are raw images, TSWC, CDnCNN, CBDNet, and NCTBNet-B. Frog and dog image source credit: Anaya & Barbu (2014).

Figure 8 Denoising results of thermal infrared images.

(A) Raw noisy images; (B) denoised by NCTBNet-B; (C) raw noisy images; (D) denoised by NCTBNet-B.

We selected several real images with different noise intensity as test samples. It can be seen from Fig. 7 that: (1) the noise range of real images is limited by the contrast denoising device, but residual noise still exists in some images. (2) For images with different noise intensity, there are two phenomena in the de-noising machine: excessive smoothing and noise residue, which is caused by inaccurate noise level estimation. However, compared with the noise residual, the raters tend to over-smooth and give a higher score. (3) The proposed models can handle realistic noise well and obtain the highest scores. As can be seen from Fig. 8, the proposed model still has a good de-noising effect on the noisy images of real thermal infrared switch cabinet.

The computation time is also a significant index to evaluate the computational complexity. We choose three different size images, and the computation time of each model can be seen in Table 3, where our model is more competitive than other denoisers.

Table 3 Running time of each denoiser (second).

	BM3D	TNRD	DnCNN	BRDNet	N2V	GCDN	BUIFD	NCTBNet	
Device	CPU	GPU	GPU	GPU	GPU	GPU	GPU	GPU	
256×256	0.66	0.011	0.012	0.036	0.013	0.033	0.045	0.019	
512×512	3.04	0.031	0.034	0.087	0.137	0.136	0.098	0.065	
1,024×1,024	11.11	0.102	0.122	0.247	0.134	0.321	0.345	0.141	

We design an ablation study to examine the contributions of each structure and NSCT. We test models on BSD68 with 25 noise level. The results are shown in Table 4. It can be easily found that both structure and NSCT are useful to our model.

Table 4 Ablation study on BSD68 with noise level 25.

	NSCT	Upper network	Down network	PSNR	
Architecture	√	×	√	28.67	
√	√	×	28.73	
×	√	√	29.12	
	√	√	√	29.39	

Conclusions

In this article, a dual convolutional neural network based on non-subsampled contourlet transformation is proposed to realize image denoising. The model connects two U-Nets in parallel, which improves the denoising performance while reducing the depth and computational cost, enables the model to retain the details of the image, and makes the model easier to train. The use of NSCT not only expands the size of the training dataset, but also avoids the grid effect, so that the network model can learn more features of noisy images. A large number of denoising experiments show that NCTBNet is superior to other most advanced denoising models in image denoising, and has good adaptability to real images and thermal infrared images.

Supplemental Information

Supplemental Information 1 Code.

Additional Information and Declarations

Competing Interests

Author Contributions

Data Availability

Zhendong Xu, Hongdan Zhao, Yu Zheng, Hongbo Guo and Shengyang Li are employed by State Grib Jilin Electric Power Co., Ltd, Liaoyuan Power Supply Company.

Zhendong Xu conceived and designed the experiments, performed the experiments, analyzed the data, prepared figures and/or tables, authored or reviewed drafts of the article, and approved the final draft.

Hongdan Zhao conceived and designed the experiments, performed the experiments, prepared figures and/or tables, and approved the final draft.

Yu Zheng analyzed the data, authored or reviewed drafts of the article, and approved the final draft.

Hongbo Guo performed the computation work, prepared figures and/or tables, authored or reviewed drafts of the article, and approved the final draft.

Shengyang Li performed the experiments, analyzed the data, prepared figures and/or tables, and approved the final draft.

Zhiyu Lyu performed the experiments, performed the computation work, prepared figures and/or tables, authored or reviewed drafts of the article, and approved the final draft.

The following information was supplied regarding data availability:

The imageNet dataset is available at https://www.image-net.org/.

The Waterloo Exploration Dataset is available at https://ece.uwaterloo.ca/~k29ma/exploration/.

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
