# Peer review of "A dual nonsubsampled contourlet network for synthesis images and infrared thermal images denoising"

_PeerJ Computer Science, doi:10.7717/peerj-cs.1817_

## Round 0.1 · original submission · Major Revisions

Dear Authors,

Your paper has been revised. Based on the reviewers’ reports, your manuscript needs major revisions before it is published on PEERJ Computer Science.

Although the methodology and results you have presented in your study seem promising, the objectives could be more explicit, and the presentation should be improved. Also, the literature review should be enhanced, thus comparing your study better to the recent related works on the subject.

In particular, a reviewer pointed out that your paper suffers from numerous unclear expressions and logical errors that need to be fixed in the revised version of your article.

Finally, I advise the authors to elucidate better the validity of the findings discussed in their study by answering all the questions posed by reviewers.

Reviewer 1 has suggested that you cite specific references. You are welcome to add it/them if you believe they are relevant. However, you are not required to include these citations, and if you do not include them, this will not influence my decision.

Reviewer 1 ·

Basic reporting

The authors proposed a dual convolutional neural network model based on nonsubsampled contourlet transform for image de-nosing. While the methodology and results look promising, the objectives are unclear, and the presentation should be improved. In addition, the literature review should be improved. For example, the following two suggested articles can be discussed as well. Please see below my comments.

Line 92. Other deep learning methods for signal and image denoising can also be mentioned here, such as Long Short-Term Memory (LSTM), and nonconvex low rank model and TV regularization:
https://doi.org/10.1109/JSEN.2023.3237383
https://doi.org/10.3390/app13127184
The first article above uses LSTM network to remove artifacts from signals, the method of which is also applicable to thermal image denoising. The second article use some of the images that you used in the manuscript and effectively denoised them. Therefore, please compare your results with the results of the second article that I mentioned above.

A discussion section should be added where you can elaborate on the advantages and limitations of each method including their computational complexity and in the light of similar studies like the second article mentioned above.

Experimental design

Abstract. Line 18. As the authors wrote here the objective of this study is to use infrared thermal imaging technology for temperature measurement utilizing infrared thermal images to find the electrical switch cabinet fault. Thus, an effective de-nosing such images could be helpful. However, Figures 5,7,8,9,etc. are general images that are not related to the objective. The authors should either revise the abstract and objectives or analyze infrared thermal images and/or include more thermal imagery for electrical switch cabinet fault.

Line 22. In the abstract. To optimize what?

Validity of the findings

Also, how do we know which de-noise method is better? What is the ground-truth image? You can start by artificially introducing a noise to a clean image and try different methods. This also refers to Lines 242 to 245.

Additional comments

The figure numbers do not follow the order in the text. For example, Figure 8 is marked as Figure 11 that is confusing. Figure 5 (d), (e), (f) are given, but I do not see panels (a), (b), (c). they appeared in Figure 9!
Please insert all the panels for each individual scene into one figure and use labels (a), (b), etc. in the top left corner of each figure. This was you have a jpeg file for each scene that are labelled and then in the caption you can easily describe which one is which. Also, please insert ground truth (noise-free) for at least some examples, so the reader can visualize which method performs better. In fact, you may display the residual image that is the denoised image minus the ground truth.

Line 248. In which study? By others? Please be clear.

Equation (1). Please define all the parameters.

Line 54. What is AWGN? Please define all the acronyms the first time they appeared.

Reviewer 2 ·

Basic reporting

This paper introduces a denoising model that combines dual convolutional neural networks, which enhances denoising performance to some extent while maintaining efficiency. It offers some novel insights into addressing real and synthetic noise issues. However, the whole paper suffers from numerous unclear expressions and logical errors. It is recommended that the authors thoroughly revise the article to improve its quality.

Experimental design

(1)It is recommended that in the comparative experiment in Table 1, the experiments conducted in the past three years be selected as references, while earlier methods such as "BM3D" and "TNRD" be excluded. This ensures that the comparative experiment is more timely and persuasive.
(2)In the experiment conducted on the BSD68 dataset, when the sigma value is 15, the GCDN model achieves significantly better results than the author's proposed model. If this is the case, what is the significance of the author's work?
(3)In the paper, only the Peak Signal-to-Noise Ratio (PSNR) objective evaluation metric is presented in the experiments. It is recommended to include other objective evaluation metrics such as the Structural Similarity Index (SSIM), Entropy, and Fourier Spectrum Loss to provide a more comprehensive assessment of the results.

Validity of the findings

(1)Why did the authors use two structurally identical U-Net models? What advantages does this have compared to using a single U-Net? The motivation behind the improvements made by the authors in their design has not been explained. It is recommended that the authors conduct ablation experiments to demonstrate the theoretical and engineering value of their model.
(2)The article makes improvements to the NCST and inverse NCST for the dual U-Net model, but these improvements appear to have limited innovation. In this paper, the authors should explain why they chose to combine networks with relatively lower performance.

Additional comments

(1)It is recommended that the authors include recent references to emphasize the importance of infrared thermography in detecting electrical switchgear faults in the past few years.
(2)In the "Introduction" section, it is recommended to add a description of deep learning methods in the denoising field, emphasizing the results achieved by scholars using deep learning methods. This will help highlight the importance of deep learning in image denoising.
(3)"Figure 4" does not correspond to the description in lines 151-155 regarding the "NSFB structure." There is an inconsistency between the figure and the textual description.

---

## Round 0.2 · Minor Revisions

Dear Authors,

Your paper has been revised. Although a reviewer has considered your paper suitable for publication, the other one thinks that your paper needs some improvements. More precisely, several statements you have used in your study must be better explained or referenced so the reader can understand their significance more profoundly.

Thank you for your interest in PEERJ Computer Science.

Reviewer 2 ·

Basic reporting

no comment.

Experimental design

no comment.

Validity of the findings

no comment.

Additional comments

The authors have revised the paper according to the comments of the reviewers.

·

Basic reporting

The adequate introduction has not been provided.

Experimental design

The details of experimental design are missing in the paper.

Validity of the findings

The data analysis has not been adequately done

Additional comments

Other similar methods for denoising thermal images may also be discussed in the paper.

The paper lacks mathematical support to the proposed method. relevant mathematical equations may be included.

---

## Round 0.3 · accepted · Accept

Dear Authors:

Your manuscript entitled "A dual nonsubsampled contourlet network for synthesis images and infrared thermal images denoising - has been Accepted for publication." has been accepted for publication in PEERJ Computer Science. 

Thank you for your fine contribution.